# Key Guidelines in Developing a Pre-Emptive COVID-19 Vaccination Uptake Promotion Strategy

**DOI:** 10.3390/ijerph17165893

**Published:** 2020-08-13

**Authors:** Jeff French, Sameer Deshpande, William Evans, Rafael Obregon

**Affiliations:** 1Strategic Social Marketing Ltd, Liphook, Conford GU307QW, UK; 2University of Brighton, Brighton BN2 4AT, UK; 3Social Marketing @ Griffith, Griffith University, Nathan Queensland 4111, Australia; s.deshpande@griffith.edu.au; 4Milken Institute School of Public Health, The George Washington University, Washington, DC 20052; wdevans@gwu.edu; 5UNICEF, Asunción PY 1806, Paraguay; robregon@unicef.org

**Keywords:** Covid-19, vaccine uptake, vaccine hesitancy, behaviour change, social marketing, communication

## Abstract

This paper makes the case for immediate planning for a COVID-19 vaccination uptake strategy in advance of vaccine availability for two reasons: first, the need to build a consensus about the order in which groups of the population will get access to the vaccine; second, to reduce any fear and concerns that exist in relation to vaccination and to create demand for vaccines. A key part of this strategy is to counter the anti-vaccination movement that is already promoting hesitancy and resistance. Since the beginning of the COVID-19 pandemic there has been a tsunami of misinformation and conspiracy theories that have the potential to reduce vaccine uptake. To make matters worse, sections of populations in many countries display low trust in governments and official information about the pandemic and how the officials are tackling it. This paper aims to set out in short form critical guidelines that governments and regional bodies should take to enhance the impact of a COVID-19 vaccination strategy. We base our recommendations on a review of existing best practice guidance. This paper aims to assist those responsible for promoting COVID-19 vaccine uptake to digest the mass of guidance that exists and formulate an effective locally relevant strategy. A summary of key guidelines is presented based on best practice guidance.

## 1. Introduction

As we work to develop a range of safe and effective COVID-19 vaccinations, the anti-vaccination movement has already fired the first shots in what will be a global public health battle. Research shows that general vaccine hesitancy (i.e., ‘the delay in acceptance or refusal of vaccines despite the availability of vaccination services’) is rising for several diseases, resulting in serious disease outbreaks. For example, 11 European countries experienced more than 1000 cases of measles in 2008 [1]. Vaccine hesitancy has also steadily increased in more than 90% of countries since 2014 [2]. Given the potential to undermine vaccination coverage, all states must take steps to understand the extent and nature of hesitancy and to start promoting COVID-19 vaccine uptake. As the WHO recommends, ‘each country should develop a strategy to increase acceptance and demand for vaccination’ [1]. Each country must consider the appropriate time to start promoting the uptake of COVID-19 vaccines based on its specific trajectory of COVID-19 infection and its ability to provide access to vaccination. As COVID-19 vaccination uptake develops, governments should continue to promote other protective behaviors such as handwashing and physical distancing.

### 1.1. The Rationale for This Paper

This paper aims to set guidelines that governments and regional bodies across the world should take to enhance the impact of their pro-vaccination strategy. We base our summary on recommended best practice with the aim of assisting professionals to digest the mass of guidance that exists in the hope that the summary contained will inform the guidelines needed to maximize uptake of COVID-19 vaccines.

It is imperative that planning for a COVID-19 vaccination uptake strategy begins in advance of vaccine availability for two reasons. First, countries will need to determine population sub-groups and build a consensus about the order in which these will get access to the vaccine. Second, we should reduce fear and concern and create demand for vaccines. A key part of this strategy is to counter the anti-vaccination movement that is already promoting hesitancy and resistance.

Since the beginning of the COVID-19 pandemic, we have witnessed a tsunami of misinformation and conspiracy theories that have the potential to reduce vaccine uptake. To make matters worse, sections of populations in many countries display low trust in governments, official information about the pandemic, and the official approach in tackling the epidemic.

### 1.2. What This Paper Does

The WHO advocates a pre-emptive pro-vaccination strategy that psychologically inoculates the population and maximizes uptake of vaccines as they become available [1]. This paper sets out the core elements of such a strategy. The paper explores key issues that relevant organizations must address and summarizes best practices that should be addressed when developing behavioral influence strategies to promote the uptake of COVID-19 vaccines effectively, efficiently, and ethically as they become available.

### 1.3. What This paper Does not Do

This paper does not set out a full review or commentary on the thousands of scientific papers and national and international guidance documents that already exist with respect to promoting vaccine uptake and reducing vaccine hesitancy. The volume and dispersed nature of this literature is, in some ways, an impediment to action as few people will have a full grasp of the multiple fields of research that inform it.

The paper also does not attempt to set out a full planning model or a ‘how-to’ guide, as numerous well-tested examples already exist [3,4,5,6].

The paper does not provide a comprehensive set of references; instead, it cites select evidence summaries and guidance documents to aid further reading.

Finally, we have not included a separate evaluation strategy, as each of the key guidelines will need an integrated monitoring and evaluation strategy to enable continuous improvement.

### 1.4. Key Guidelines to Develop a Pre-Emptive COVID-19 Vaccination Strategy

Context matters. Each government and public health service face its own set of unique challenges. Different countries also have differing resources, capacities, capabilities, assets, and constraints. Regardless of such settings and challenges, governments and relevant bodies can action a number of key processes identified in the literature that will enhance vaccine uptake. We set out these key action areas in the guidelines below. See Table 1.

It is highly likely that in the coming months the WHO and other public health institutions will issue guidance about how to optimize the uptake of COVID-19 vaccines. We present the guidelines set out in this paper as an ideal model based on the lessons learned from successful intervention programs to inform such guidance. Organizations, however, should approach each action area in a locally relevant way. It is also clearly a big ask to address all the recommended guidelines identified, but the more of these actions that can be applied, the more likely it is that a successful uptake strategy will be delivered.

## 2. Behavior Change Planning

It is important that a systematic approach to planning is adopted. There are numerous planning models from the fields of health promotion and social marketing that authorities can use to define objectives, design processes, and conduct monitoring and evaluation of efforts to promote vaccine uptake [5]. The most crucial action is to set out a transparent (Open access) and a logical plan that covers all the essential components contained in the guidelines included in this paper. However, a coordinated and a systematic approach will require strong leadership.

Behaviour change plans should also be informed by lessons from the fields of management, logistics, and emergency and disaster planning such as the Highlight, Audience, Behaviour, Intervention, Test (HABIT) behaviour disaster change planning framework [4,7].

Authorities should also consider lessons and tips set out in several detailed planning models and guides developed specifically for vaccine promotion efforts such as:✓**WHO.** Guide to Tailoring Immunization Programs (TIP) for infant and child vaccination [1]. The TIP principles apply to communicable, non-communicable, and emergency planning where behavioral decisions influence outcomes [8] https://www.who.int/immunization/programmes_systems/Global_TIP_overview_July2018.pdf?ua=1✓**European Centre for Disease Control** (ECDC). Technical Guide to Social Marketing https://www.ecdc.europa.eu/en/publications-data/social-marketing-guide-public-health-programme-managers-and-practitioners✓**WHO.** Improving vaccination demand and addressing hesitancy. https://www.who.int/immunization/programmes_systems/vaccine_hesitancy/en/✓**ECOM:** Effective Communication in Outbreak Management (ECOM) [9]. The E.U. funded ECOM project brings together multiple disciplines to develop an evidence-based behavioral and communication package for health professionals and agencies throughout Europe in case of significant outbreaks of infectious diseases. http://ecomeu.info/✓**Tell Me.** Review of population behavior and communication during pandemics: https://www.tellmeproject.eu/✓**Human Center Design for Health**. A comprehensive set of tools developed by UNICEF to apply the human-centered design approach to challenges facing health services, with a particular emphasis on demand for immunization and health services. https://www.hcd4health.org/resources✓**Social Science Research for Vaccine Deployment in Epidemic Outbreaks**. A practical guide to using social science research and insights to better understand social, behavioral, cultural, community and political dynamics as part of efforts to introduce vaccines in epidemic outbreak settings. https://opendocs.ids.ac.uk/opendocs/bitstream/handle/20.500.12413/15431/PracApproach%206.pdf?sequence=2&isAllowed=y

Further generic planning guidance can be found at:✓**Building Better Health: A Handbook for Behavioral Change.** “The Handbook blends proven disease prevention practices and behavioral science principles into a one-of-a-kind, hands-on manual.” [10] (p. xiii).✓**CDCYNERGY Planning Tool for Social Marketing**. Centers for Disease Control and Prevention planning tool for social marketing, Atlanta, GA. Also available is CDCynergy “Lite”, intended for those who have previous social marketing experience and those who are familiar with the full CDCYNERGY edition. https://www.thecommunityguide.org/resources/cdcynergy✓**Applying Behavioral Insights—Simple Ways to Improve Health Outcomes**. A tool for the application of behavioral insights to improving health outcomes from the World Innovation Summit for Health.✓https://www.imperial.ac.uk/media/imperial-college/institute-of-global-health-innovation/Behavioral_Insights_Report-(1).pdf

## 3. Audience Targeting and Segmentation Strategy

If governments develop vaccine uptake programs based only on expert opinion, they are likely to be suboptimal [11,12]. What is required is an approach that seeks to gather as much understanding as possible about what people say will prevent, encourage, and assist them in taking up vaccines. Authorities must understand what people value and what they fear when developing an effective promotional program.

A targeted approach that uses a different intervention mix for different subsets of the population will be more effective. People do not respond uniformly to preventive interventions. For example, being older, female, and more educated is associated with a higher likelihood of adopting protective behaviors [13,14].

‘Insight’ data about citizens’ attitudes, beliefs, wants, and behaviors should inform interventions. Insights are ‘deep truths’ and understanding about why people act as they do. Such insights can be developed from formative qualitative and quantitative survey research, observational data, demographic data, service use data, problem or issue tracking data, and epidemiological data. The development of deep insights into people’s lives, with a focus on what will or will not motivate or enable people to take up vaccination, is a crucial investment that must be made to inform all aspects of vaccination promotion uptake strategy.

A key component of behavioral planning is the setting of measurable behavioral objectives that are relevant and timely in relation to maximizing vaccine uptake. Setting measurable goals related to uptake, attitudes, intention, understanding and beliefs will help focus behavioral planning and enable meaningful ongoing tracking and evaluation of impact [15].

Segmentation is key to success. Segmentation is the identification of groups who share similar beliefs, attitudes and behavioral patterns. Segmentation goes beyond demographic, epidemiological, and service uptake-based targeting. Segmentation includes data about people’s attitudes, values, understanding and observed behaviors. Population segmentation models enable public heath planners to tailor interventions to specific audiences [16]. Fournet et al. have identified four unprotected and under-protected population groups that could form the basis for the development of a locally developed strategy [17]:‘The hesitant’–Those who have concerns about perceived safety issues and are unsure about needs, procedures and timings for immunizing.‘The unconcerned’–Those who consider immunization a low priority and see no real perceived risk of vaccine-preventable diseases.‘The poorly reached’–Those who have limited or difficult access to services, related to social exclusion, poverty and, in the case of more integrated and affluent populations, factors related to convenience.‘The active resisters’–Those for whom personal, cultural, or religious beliefs discourage them from vaccinating.

Other segments that need dedicated foci are health and social care workers. Studies have revealed that certain healthcare workers hesitate to vaccinate themselves or their family members [9,18]. The ECDC provides guides and toolkits for healthcare workers, immunization program managers, and public health experts, to support their efforts in addressing vaccine hesitancy [19].

Frontline workers can be a significant source of trusted advice and information but are often not optimally used in such roles. These workers lack training and support in advocacy roles and may also lack a full awareness of risks and safety issues associated with the disease and vaccination. Governments and responsible agencies should facilitate support structures that increase worker awareness and willingness to act as public health advocates.

## 4. Competition Strategy

To effectively promote and maintain demand for a COVID-19 vaccine, governments and regional bodies must develop an insight-informed pro-vaccination strategy that includes action to reduce the impact of four kinds of competition:Active competition from the ani-vaccination movementPassive competition in the form of inaccurate media coverageCompetition from negative social normsCompetition in the form of structural and economic factors

### 4.1. Active Competition from the Anti-Vaccination Movement

Effective campaigning against vaccine misinformation should focus on the dangers of the disease as well as on the benefits of the vaccines, which can include highlighting protection. Such approaches draw on the powerful motivator of fear of loss along with the possibility of gain of positive health [20]. Intervention designers should involve the target populations in building campaigns, and use data-supported insights about what will and what will not motivate them to take up vaccine programs and about how to frame the promotion of vaccination. A competition strategy that seeks to reduce the impact of those promoting hesitancy that emphasizes fact-checking and myth-busting may do more harm than good. Such approaches often repeat misinformation as part of rebuttal strategies.

Engaging directly with conspiracies often spreads rather than closes down such views. People often exhibit what Lord calls confirmation bias; they look and accept information that fits with their existing views and reject information that runs counter to their existing views [21]. So, when repeating misinformation in order to debunk it, people may just hear the misinformation. A more effective approach is a combination of positive messaging that emphasizes the protective (individual, family, and community) benefits of the vaccine and the loss associated with not being vaccinated (death, poor health, loss of freedom and social solidarity, inability to travel, etc.) [22,23].

### 4.2. Passive Competition in the Form of Inaccurate Media Coverage

Anti-vaccination advocates should not be left free to spread misinformation. Public health authorities and their coalition partners, including both the traditional and digital media sectors, should proactively work together to reduce and remove at speed false content and misleading information. Traditional media providers should be supported and briefed so that they are aware of anti-vaccination propaganda identified by public health authorities and do not repeat it.

Traditional media and social media sectors should also provide authorities with the information they have detected that anti-vaccination advocates are propagating so that information can be rebutted. Public health agencies should seek protocols with media providers about the issue of how journalistic balance will be addressed. Agreements should be put in place about how the media will identify and flag false and misleading anti-vaccination information and advocates. In this regard authorities and media channel providers should be alert to ‘Astroturfing’ (anti-vaccination advocates disguising their views as coming from grass roots movements) and act swiftly to expose such tactics. Finally, agreements should be developed about how and when misleading information and advocates of such information should be removed and flagged as being problematic on social media.

### 4.3. Competition from Negative Social Norms

Distrust in elites and experts and political populism can also fuel antivaccination sentiment [24]. Social norms and cultural influences can have a significant effect on people’s willingness at the population level to take up vaccine programs [25]. As an initial step, authorities need to understand what informs social norms and beliefs. Persuasive efforts should appeal to the values and beliefs that people already hold, such as a desire to protect family members, rather than a focus on factual or probabilistic messaging.

Validating people’s existing motivations and using them to encourage behaviour is more effective than trying to shift people’s world view. If, however, people hold incorrect opinions about the social norms prevailing in their community, for example, the erroneous belief that most people oppose vaccination, it can be helpful to inform them that a high percentage of people do in fact, support vaccination. Subjective social norms, i.e., those that are informed by what we think key others in our social circles believe, are also crucial in promoting vaccine uptake [18,26,27].

Opinion leaders in the anti-vaccination community may hold negative attitudes and beliefs, so intervention organizers should also develop interventions with such key informants to address these concerns and seek to turn such informants into advocates for vaccination.

Previous reviews of vaccine demand campaigns using a systematic process, such as in the area of Human Papilloma Virus (HPV) vaccine, have found that myths and misinformation, often prevalent in communities, can also pose significant barriers to vaccine adoption. Evans et al. studied several HPV and cervical cancer awareness studies in low- and middle-income countries (LMICs) [28]. These studies confirm many widely reported barriers to HPV vaccination; these include myths (e.g., the vaccine causes infertility), beliefs that it will increase promiscuity, negative social norms within social groups, and concerns about safety and efficacy. Solutions to these barriers include:Increasing knowledge about the risks prevented by the vaccine.Promoting understanding that the community of interest is at risk; improving beliefs in vaccine safety, effectiveness, and community benefit.Dispelling unfounded myths.Building a social norm that vaccination uptake is widespread and accepted in society (descriptive and injunctive normative beliefs).

### 4.4. Competition in the Form of Structural and Economic Factors

Vaccine uptake strategy must address difficulties in accessing vaccines due to cost, lack of transportation to vaccination sites or clinics, and/ or a lack of a cold-chain network. Authorities need to work with partners across government, NGOs, communities, and the for-profit sector to reduce these barriers. Poor access can reduce confidence in and demand for the vaccine. Vaccine uptake promotion should thus facilitate availability and convenience.

It is vital that countries review their public health finances early on to allocate funds to vaccinate their populations, as many countries already carry large debts. To inoculate the entire global community will require significant resources. Countries with lower incomes will need to develop plans to access support from the international aid programs provided by governments, U.N. bodies and foundations, and other sources to secure adequate supplies of vaccines.

Promoters of the COVID-19 vaccine should also consider that their efforts do not negatively impact on the availability and the uptake of other vaccine programs, predominantly for children.

## 5. Mobilization

Public health organizations rarely have sufficient resource capacity to develop, deliver, and maintain population-level change-focused programs. Building and sustaining coalitions of organizations and individuals who can assist through the provision of resources, expertise, credibility and access is a crucial early action that needs to be addressed. Critical asset identification and management falls into three main categories: government capacity coordination, private sector and NGO sector mobilization, and the mobilization of civil society.

Building alliances within government and across departments is a crucial aspect of asset identification and mobilization [29]. There is a need to develop plans and structures to coordinate action between government agencies and departments and organizations such as hospitals, clinics and schools [30]. An alliance or coalition team should also coordinate mechanisms and resources and set out chains of command and responsibilities.

The NGO and private sectors can play a pivotal role in promoting the uptake of vaccines. Partnerships with the pharmaceutical industry to develop, manufacture, promote, and distribute vaccines are underway across the world. Many other for-profit organizations can also be harnessed to provide logistical and promotional support. The NGO sector is also well placed in terms of its reach, high level of understanding about local communities, and high levels of trust to act as a critical advocate and network for vaccine uptake.

The third leg of the asset and capability resource base is civil society, represented by community groups and associations such as religious groups, community associations, recreational groups and community charities and volunteers. These groups and communities can play a crucial role in encouraging vaccine uptake and assisting with distribution and access. However, the part that civic society can play in promoting and helping with vaccine uptake is highly country-specific; therefore, local plans will need to reflect the role that such groups can play [30,31,32].

Developing and maintaining a vaccine promotion coalition of government, the private sector, the NGO sector, and civic society requires resources and staff with expertise in creating and managing stakeholder relationships. Authorities need to identify the resources needed to undertake these essential tasks, set objectives, monitor progress, and provide feedback.

## 6. Vaccine Demand Strategy

Well planned, evidence-based, and theory-informed health communication and health marketing can significantly impact behavior and vaccine uptake [9,33,34]. Well-designed campaigns, together with the application of behavioral science techniques, need to be supported by ease of access to vaccines, distribution networks and logistics, and taking notice of broader socio-economic and cultural factors [35,36].

Those responsible for creating demand for the vaccine need to work with vaccine suppliers, administrators, and those delivering vaccination to bring together a full mix of demand-side and supply-side interventions. The intervention mix needs to include coordinated action in the fields of prioritization and access policy, supply systems, and promotions strategy. Prioritization is especially critical, given insufficient availability, especially after the initial months of vaccine launch. More important than building general demand are building awareness and support for COVID-19 vaccination prioritization plans and fostering high acceptance among people in priority groups. 

The key to promoting demand is a deep understanding of what will enable and encourage uptake. Campaign managers should conduct formative research including secondary research based on published literature and case studies and primary research with interviews and surveys in each population to gain audience-specific insights. Governments will need to deliver and communicate what mix of incentives and penalty interventions will be used to promote demand [37].

Demand strategy will also need to be supported by the development of a compelling, insight informed and segmented promotion that speaks to people’s needs, values, and wants. Health communicators must develop narratives that emphasize the positive personal, family, and community benefits associated with vaccine uptake. The demand strategy will need to include guidelines that reduce the influence of anti-vaccination advocates (see sections below for critical steps to consider when developing a competitor strategy). The demand strategy must also utilize positive narratives in both traditional and social media and apply behavioral influence tactics informed by behavioral sciences [10,38].

## 7. Community Engagement Strategy

The WHO recommends that every country should include ongoing community engagement and trust-building programs. Programs should be focused on confidence-building and active hesitancy prevention, together with regular national assessments of population concern and trust [1,39,40,41]. Trust is built and maintained through transparency, constancy, active listening programs, and encouraging dialogue. Agencies and governments need to share knowledge about certainty and uncertainty. Governments and public health agencies also need to pre-empt and address any safety issues that are expressed or felt by the public or media [41].

Governments should also be transparent about vaccine licensing, manufacture, and prioritization planning. Consistency of both messaging and policy directives is also crucial. The absence of these conditions will trigger confusion and reduce trust [42].

Anti-vaccination attitudes do not always relate to factors like level of education [43]. Instead, they are often related to anger and suspicion towards elites and experts and increasing support for anti-establishment political concerns. Governments should listen actively and build dialogue, encouraging continuous feedback from citizens, key commentators, and influencers. Regular proactive public media and influencer briefings should also form a central plank of trust-building strategy. The application of citizen-focused and human-centered design principles can also enhance program development and implementation [44].

## 8. Vaccine Access Strategy

Relevant agencies should realize the need for a coordinated mix of interventions to promote vaccine access, led by a strong leadership team [45]. Promoting uptake through the media and community advocates is a critical element of any pro-vaccination strategy, but it is not a panacea for convincing everyone reluctant to vaccinate. Research shows that behavioral change is a complex process that entails more than having adequate knowledge about an issue. Uptake and hesitancy are also related to cultural factors, attitudes, motivations and experiences, social norms, and structural barriers. Understanding the multiple factors involved in people’s decisions is, therefore, key to success. Governments and public health authorities can enhance the effectiveness of their efforts by combining multiple strategies [46]. For example, they could integrate financial and non-financial incentives, call and reminder interventions, along with penalties for non-compliance by imposing restrictions on travel, education, or employment [37].

Vaccine access information, requirements and support will need to reflect each country’s vaccination implementation strategy. Will it be mandatory? Will there be penalties for non-compliance? Communicators should deliver implementation and access strategies through a segmented approach that provides specific and relatable information to identified subgroups of the population about how and when they can have access to vaccination. Call mechanisms will need to be established and monitored as part of this element of the strategy.

With regard to vaccine selection, assuming that the medical fraternity has developed several safe and effective vaccines by 2021, governments and public health authorities will need to explain to the population why they selected a particular vaccine in terms of its efficacy, safety, cost, etc.

Authorities will also need to explain their reasoning for the prioritization model for the vaccination that they adopt. For example, if a risk-based approach is adopted in which older people and care workers are prioritized over younger people and non-essential workers, this needs to be explained. Governments and regional bodies need to explain and justify these decisions in terms of health protection, social and economic imperatives, safety and cost imperatives. Schedules and timetables for total population vaccination should also be developed and shared before vaccination roll out begins so that everyone understands when they will get access. Ideally authorities should share their plans for vaccine roll out prior to availability so that there is time for ethical and procedural issues to be publicly debated and a consensus reached.

## 9. Marketing Promotions Strategy

A coordinated national approach to communication will be successful among many groups, but not all [37]. Success depends on the nature and degree of immunization hesitancy and the degree of segmentation. Tailored messages focusing on known motivators for specific groups are more likely to produce a desired behavioral response than a ‘one size fits all’ approach [47,48,49]. To produce tailored messages, we recommend quantitative and qualitative formative research and ascertaining the efficacy of strategies with pre-test research before launch.

As stated previously, there is a need to set out a compelling narrative that avoids ‘backfire effects’ [50], validates people’s concerns, and addresses both fear of loss and the positive gain that will accrue from vaccine uptake. As Tversky and Kahneman have demonstrated, when confronted with choices we are averse to any that might result in perceived loss [51]. We also do not like being confronted with complex choices. It follows that, if governments want to influence people to take up vaccination, they are more likely to be successful if the strategy emphasizes the positive gains accrued from vaccination, the loss that will occur if vaccination is refused, and that access to vaccines is easy.

We know that the perceived attractiveness of options varies when communicators frame the same choice differently. Therefore, the language used, the imagery, the messengers, and audio-visual effects are all important considerations that communicators should pilot test. As stated previously, authorities should tailor their promotional strategies by subgroups of the population, as each segment will respond differently to varied messaging and narratives.

Familiarity and trust in the messenger, as well as the message, is also a crucial success feature in tackling vaccine hesitancy [1,52]. Authorities should determine which campaign face and voice should be used based on formative research with the target audience. Messages that come from a variety of trusted sources are likely to make a vaccine promotion programs more successful. Spokespeople recruited from trusted groups, including healthcare professionals and relatable members of the public, can enhance the effectiveness of campaigns. High-profile personalities can also be effective in communicating messages, as they lend their prestige and trust to the health communication activity. The use of religious leaders (like the cooperation offered by Muslim religious leaders in India to communicate the importance of polio vaccination), community influencers and third-party advocates, such as teachers, can also improve support for vaccination uptake [53].

As part of long-term public health strategy, governments and public health agencies should enhance media and digital literacy in schools and community settings, specifically related to health and vaccine topics [54]. Newly acquired literacy will equip the public to identify reliable sources of information and encourage reporting of misinformation to social media providers and regulating authorities.

## 10. News Media Relations and Outreach

The news and general media can contribute significantly to address fears and risk perceptions, which can hurt vaccine uptake [55]. It is, therefore, necessary to develop a proactive strategy for working with traditional media. Any media management and engagement strategy that is developed will need to include proactive, rolling media briefings, story generation, editorial feeds, facilitating access to medical and other clinical and public health experts, advisers, and data. The media management and engagement strategy will also need to include 24/7 media monitoring and rebuttal/correction systems.

Communicators should mediate ongoing relationships between media contacts and experts who can provide accurate opinions on all aspects of vaccine promotion and safety. Authorities should additionally monitor the strength of this relationship and address rapidly any conflicts that may arise. The responsibility of government agencies and others advocating for COVID-19 vaccination is to communicate better, more visible, and more highly credible messages than the sceptics.

Successful media engagement is more likely when the public health system has developed a strong collaborative and open relationship with key editors, sub-editors and journalists. Public health authorities and governments should continuously nurture trust and positive working relationships with media organizations so that the audience views the former as accessible and trustworthy. This will, however, require government authorities to be transparent, honest, and open regarding vaccine safety and effectiveness data that could be, or is, worrisome.

## 11. Digital Media

Anti-vaccination advocates abound on Facebook, Twitter, WhatsApp, and YouTube. Social media platforms are already buzzing with misinformation about COVID-19 vaccine safety, development, and planned rollout, months before vaccines are ready to be used at population level. It is encouraging to see such media platform owners starting to act against the anti-vaccination movement. For example, Instagram avoids health misinformation in its Explore page; YouTube has demonetized anti-vaccination videos and GoFundMe has recently taken down anti-vaccination fundraising appeals. Governments and their public health agencies need to develop a dialogue and joint strategy with social media platform providers to review and action against anti-vaccination misinformation and vaccine hesitancy promotion. Governments and regional bodies should convince or regulate platform providers to remove misinformation.

Public health authorities need to build a proactive COVID-19 vaccine trust capacity for active engagement in the social media space as part of their overall promotional strategy [56]. Social media platforms are now the primary information source and communication channel for a large and growing number of citizens. Public health agencies need to invest in building teams of specialist staff trained and capable of understanding how to build and maintain social media presence.

The key responsibilities of public health staff focused on social media are the development of and support for continuous positive story streams, nurturing multiple supportive voices, and amplification of pro-vaccination grassroots advocates. These dedicated staff need to support pro-vaccine influencers, advocates and social networks. Public health staff can also assist in the identification of and responses to false social media posts. The team should address such negative posts instantly to prevent the decline of trust in public health authorities. We know, for example, that parents who are resistant to getting their children vaccinated are more likely to have based their decision on information obtained on the internet [57].

## 12. Conclusions

The strategic and tactical guidance set out above provides a framework for promoting the uptake of COVID-19 vaccines as they become available. This paper also acknowledges the importance of evidence and theory-driven behaviour change tools in addressing vaccine hesitancy. This is consistent with WHO’s recent establishment of the Technical Advisory Group on Behavioral Insights and Sciences for Health [58]. Key to the success of promoting vaccine uptake will be a significant and sustained strategic program, including strengthening of local capacities, to build and maintain confidence and trust [59]. A crucial factor in the delivery of such a trust-building and demand building approach is the need for investment in communication, behavioral influence, and community engagement capacity and capability. Communication and behavioral influence are often underfunded or under-resourced in public health organizations and within government ministries. Building communication and behavioral influence capacity and expertise should be a priority. It is now often said that everything will be different in the post COVID world; hopefully one difference will be a commitment to investment in developing and delivering the key action elements set out in this paper. This investment will need to be sustained over time in line with best practice requirements regarding risk communication and community engagement so that we are better prepared for inevitable future events [39]. The authors acknowledge that countries, high-, low- and middle-income, have been using many of the guidelines described in this manuscript to foster high vaccination coverage. The challenges are not that they are unaware of the actions described here but rather: (1) they have very limited resources (e.g., money, people) to implement all the actions at the scale the authors are recommending; and (2) they are responsible for promoting and achieving compliance with vaccination schedules, not just a single vaccine. Governments and relevant bodies should bear these limitations in mind as they consider our guidelines.

## Figures and Tables

**Table 1 ijerph-17-05893-t001:** Key guidelines for Developing a Proactive COVID-19 Pro-Vaccination Strategy.

Key Guidelines	Guideline Completed	Guideline Underway	Guideline
Not Completed
1Behavior change planning			
2Audience targeting and segmentation			
3Competition and barrier analysis and action			
4Mobilization			
5Vaccine demand building			
6Community engagement			
7Vaccine access			
8Marketing promotions strategy			
9News media relations and outreach			
10Digital media strategy

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
