# Peer review of "Key Guidelines in Developing a Pre-Emptive COVID-19 Vaccination Uptake Promotion Strategy"

_ijerph, 2020, doi:10.3390/ijerph17165893_

Round 1

Reviewer 1 Report

The author affiliations in line 4 are wrong. The last author does not have an affiliation and numbers 3 and 4 are not used at all. Consider rephrasing line 59. References should be placed at the end of each sentence rather that the middle. In line 150 No 2 should be before the full stop not after. In lines 430-434 the Authors contributions, funding, acknowledgment and conflict of interest sections are blank suggesting a rather careless approach at the time of submission. In the Reference section, some references end with a full stop while other are not. There are many internet links without data about when those webpages where cited. 

Author Response

Please see the attached letter and revised manuscript. Thank you.

Reviewer 2 Report

  • Concerned about frequent reference to "the royal we", i.e., page 2, line 49. Does the "we" refer to policy makers and/or scientists? One of the challenges has been a coordinated effort. Authors should identify their identified audience for this manuscript within the manuscript.  Many of the recommendations are outside the power of the readership of this journal. 
  • Three reasons are stated: 1) consensus, 2) reduce fear and 3) create demand (page 2, lines 49-51)
  • Identification of vaccine - what challenges will be created if the vaccine is not available to all due to insufficient amount? 
  • page 4, line 131-132, what strategies are proposed to gather information and develop an understanding in vaccine uptake?
  • page 4, lines 139-145, insight data about attitudes, beliefs, wants and behaviors are not gleaned from observational data, demographic data... or epidemiological data.  Gaining "Insight" would require questionnaires or qualitative methods. "Insight" is critical; the absence of guidance at this point in the manuscript is a considerable weakness.
  • page 5, lines 183 and 185 need citations.  These statements do not have universal understanding. 
  • page 5, Planning and implementation strategy and page 9, line 387 - these sections do not address that a coordinated/systematic approach will requires leadership.  This recommendation would not be refuted, but education of leadership is missing.
  • page 6, line 228  and page 7, line 299 - again the vague treatment of relevant authorities weaken the reader's understanding of who is positioned to developed trusted, data driven strategies?  
  • page 7, line 313 - what is the means to tailored messages?  given the importance of grounding messages in beliefs and perception, the lack of detail in "insight" and tailored messages, leaves the reader ill-equipped to act. 

Author Response

(The authors gave the same response as above.)

Reviewer 3 Report

This manuscript makes the case for immediate communications planning for a COVID-19 vaccination uptake strategy. It is argued that such an effort is needed to build consensus about which groups should get access to the first available doses and to create vaccination demand. The manuscript also notes that anti-vaccination groups and individuals are currently promoting COVID-19 vaccine hesitancy and resistance. This manuscript identifies the major actions that governments and regional bodies should take to develop effective COVID-19 vaccination strategies.

Broad

The strengths of this manuscript include:

  • The manuscript has much significance. The authors call for immediate efforts to develop a COVID-19 vaccination uptake strategy is timely and appropriate. COVID-19 vaccination strategies, particularly those focused on establishing confidence in the vaccine development process and the importance of high uptake, should be undertaken while vaccines are being evaluated in clinical trials.
  • The manuscript highlights important frameworks and steps related to social marketing and behavior change communication strategies (e.g., audience targeting and segmentation, competition and barrier analysis) as well as comprehensive models and guides (e.g., WHO’s Tailoring Immunization Programmes, CDCynergy).
  • The manuscript provides much helpful information and specifics about the major domains of activities involved in developing and implementing multi-component and multi-faceted vaccination demand programs. They have identified and described the steps as well as the needed outcomes (e.g., trust) associated with vaccination acceptance.

The weaknesses of this manuscript include:

  • The key action checklist is of limited value as a checklist. Almost all of the “actions” are very broad, have a large scope, and require significant resources. Most of the “actions” are relatively large domains and many would consider each of the actions as having a distinct strategy. The domains overlap and some of the domains operationally involve actions that are listed as distinct actions. The suggested order and category titles add to the confusion. Often, for instance, the development of a social marketing or communication strategy (or in this case, a “pro-vaccination” strategy), begins with a planning process that involves a situation analysis. It is common for a situation analysis to involve asset mapping, consideration of the 4P’s (product, place, price, promotion) and competition/barrier analysis, including as part of a Strengths-Weaknesses-Opportunity-Threat (SWOT) assessment. That assessment, in turn, would inform decisions related to audience segmentation and targeting, objective setting (e.g., specific vaccination demand outcomes), and message strategy (e.g., what concepts and key themes should be highlighted to achieve communication and behavior objectives). The next steps would involve decisions related to partnerships and channels (e.g., community engagement, coalition building, traditional media use, digital media use, and promotions) and the specific objectives for each (e.g., to build trust). The final stage would be implementation or action. As such, the items listed in the checklist are really broad domains or considerations rather than a set of specific activities that more clearly would be undertaken in a specific order.
  • Related to the comments above, the titles/labels for some of the actions are problematic and would benefit from revision. “Trust building and community engagement” seems odd because “community engagement” is an action, while “trust building” is the purpose. “Vaccine demand building” is a broad domain encompassing a multitude of possible actions. “Asset mapping, mobilization & coalition building” places asset mapping in a category that includes a broad category for achieving behavior change or vaccine demand (i.e., mobilization) and a process for doing mobilization or community engagement (i.e., coalition building). “Planning and implementation” merges two distinct domains and this really isn’t a distinct step in a strategy process. A strategy process is planning and doing the many things listed in other steps would involve implementation. “Messaging and Promotions strategy” brings together two sets of activities that are often undertaken independently (though there is some overlap). “Traditional media management” signals management of an activity rather than action. It also is misleading in this manuscript in that the text related to media management is primarily about news media relations and outreach rather than using purchased or donated mass media time and space. The same is true for “digital media management.” The supporting text does not describe specific ways to utilize social media but instead offers examples of how it has been used. The text on health literacy is oddly placed in the digital media section. Finally, vaccine access is not a singular action step. It is a complicated domain involving many decisions and varies widely across countries. In some countries, private sector enterprises are heavily involved in vaccine access elements (e.g., vaccine distribution and administration) while in other countries government agencies manage much of the system.
  • Given the shortcomings in the checklist, the manuscript's organizational structure would benefit from strengthening (e.g., to be better aligned with revisions to the "checklist"). It may be helpful to drop the notion of a checklist and instead recast what are now "activities" as major domains that should factor into the development of country COVID-19 vaccination strategies.
  • The quality of the presentation needs to be strengthened. At times, the manuscript seems to be offering guidance and advice based on what is done in the U.S., while at other times, including in the introduction, it strives to be non-country specific. Given the diversity that exists across the globe when it comes to immunization programs, it would help to make clear early on who really is the target audience for the advice being offered. It should also be kept in mind that many countries, particularly high-income countries but low- and middle-income as well, have been using many of the items described in this manuscript for at least a decade to foster high vaccination coverage. The challenges are not that they are unaware of the “actions” described here but rather 1) have very limited resources (e.g., money, people) to implement all the actions at that scale the authors are recommending, 2) are responsible for promoting and achieving compliance with vaccination schedules, not just a single vaccine, and 3) do not use vaccination mandates to achieve high uptake.
  • Along with a more accurate section heading, the section on “Traditional Media Strategy” should be revised to better reflect the realities of journalism and news media. First, journalists and news media should not be considered “partners,” particularly for governments and government agencies. We are all better served if they are viewed as independent and need government “watchdogs.” In the case of new, never before used vaccines, we all benefit from having news media and journalists challenging government and medical assertions about the safety and effectiveness of new COVID-19 vaccines. Hopefully all the licensed vaccines will be safe and effective, but history shows that isn’t always the case. It is good advice to establish good relationships with journalists and news media, but that should mean government transparency, honesty, and openness. Yes, this involves being proactive, but it better framed as having a true and extended commitment to transparency, honesty, and openness, including vaccine safety and effectiveness data that could be or is worrisome. Along these lines, I would drop the idea that “change specialists should ensure that journalists practice informed balanced reporting. . .” One, neither specialists nor government employee can ensure that. Two, they shouldn’t see “ensuring” such an outcome as one of their responsibilities. And three, anti-vaccination advocates will be present and visible regardless of news media coverage. The responsibility of government agencies and others advocating for COVID-19 vaccination is to have better, more visible, and more highly credible messages than the skeptics.
  • The authors should keep in mind that while it is important to build demand for COVID-19 vaccines and vaccination, the initial supplies of recommended vaccines will be very limited in all countries. This is likely to be the case for one or two years, perhaps longer. More important than building general demand are: 1) building awareness and support for COVID-19 vaccination prioritization plans and 2) fostering high acceptance among people in priority groups.

Specific comments

  • Lines 29-35: Immunization coverage data in many countries has not shown that vaccine hesitancy “is rising for most diseases resulting in alarming worldwide figures and a number of serious disease outbreaks.” While it is true the number of vaccine preventable disease outbreaks has increased in recent years, the primary increase involves measles and mumps. Outbreaks involving polio have happened but most are linked to circulation of vaccine-derived viruses. This sentence needs better precision (e.g., “alarming worldwide figures” is vague) and citations to support examples or claims.
  • In general, I would recommend not using the word “ensure.” It is not possible for government agencies to “ensure” (i.e., guarantee an outcome) unless mandates are used and enforced. For example, line 40 states that “Governments should ensure that the promotion of COVID-19 vaccination uptake does not compromise the promotion of other protective behaviors such as handwashing and physical distancing.” Based on what is known regarding influenza vaccination promotion, not only can government agencies not ensure that won’t happen, it should be expected that COVID-19 vaccination will affect promotion of handwashing and physical distancing. For example, if vaccination is considered essential for achieving herd immunity and is far more effective for preventing COVID-19 transmission, than messaging and promotion will highlight that – including by potentially saying that handwashing and physical distancing is not enough.
  • Line 340: the word “unjustified” is not necessary before fears unless a definition is provided that distinguished “unjustified” from “justified” fears.

Author Response

(The authors gave the same response as above.)

Round 2

Reviewer 3 Report

The authors have done an excellent job of responding to the reviewers' comments and suggestions. This is a much strengthened manuscript and should be of much interest and value to the journal's readers. There is much of significant value in the revised manuscript, including the resources identified in the Behavior Change planning section.

My comments with respect to further strengthening the revised manuscript are:

  • Lines 48-50: it isn't clear who "we" is and it is also the case that priority groups will likely vary by country. I would recommend rephrasing to "First, countries will need to determine population sub-groups and build a consensus about the order in which they will get access to vaccine."
  • Lines 152-153: I think the first sentence should be re-ordered to "'Insight' data about citizens' attitudes, beliefs, wants, and behaviours should inform interventions."
  • There are many references to "data-driven" strategies, but often the "data" used to guide strategies are qualitative and modest. I would thus suggest "research" or "insight"-informed strategies. 
  • Line 185 - "form" should be "from"
  • Lines 188-192: The current wording comes across as too dismissive of safety concerns. It would be better to state that effective campaigning against vaccine misinformation should focus on the benefits of vaccination (which is what is stated in lines 199-203), which can include highlighting protection against the dangers of disease. Also, I would be careful with the assumption that COVID-19 is a "probable severe threat." One, for the vast majority of people, it is not in terms of fatality or severe illness likelihood. Two, many people, for that reason, don't consider it a severe threat. 
  • Line 212 - the words "to agree" do not belong in the sentence
  • Lines 213-214: the words "and agree" do not belong in the sentence
  • Line 218: the word "of" should be deleted
  • Line 220: expert's should be experts'
  • Line 228: the notion that highlighting social norms prevailing in their community is advice that is often put forward. The problem, though, is that the community used by people when it comes to opinions and beliefs is not the community that holds different beliefs (e.g., the broader community) but rather the community that holds similar beliefs. People with "incorrect beliefs" often hold beliefs that are aligned with social norms - however it is the social norms of others holding similar beliefs.
  • Line 308 - "speak" should be "speaks" 
  • Lines 331-332: There are too many adjectives in the sentence. It appears "citizen-directed focused" should be "citizen-focused."

Author Response

We have responded to the reviewer's valuable comments. Please see the attached response to reviewer letter. Thank you.
